# Naturopathy as a Model of Prevention-Oriented, Patient-Centered Primary Care: A Disruptive Innovation in Health Care

**DOI:** 10.3390/medicina55090603

**Published:** 2019-09-18

**Authors:** Ryan Bradley, Joanna Harnett, Kieran Cooley, Erica McIntyre, Joshua Goldenberg, Jon Adams

**Affiliations:** 1Helfgott Research Institute, National University of Natural Medicine, Portland, OR 97201, USA; joshua.z.goldenberg@gmail.com; 2Department of Family Medicine and Public Health, University of California, San Diego, La Jolla, CA 92093, USA; 3University of Technology Sydney, Australian Research Center in Complementary and Integrative Medicine (ARCCIM), Ultimo NSW 2007, Australia; joanna.harnett@sydney.edu.au (J.H.); kcooley@ccnm.edu (K.C.); erica.mcintyre@uts.edu.au (E.M.); jon.adams@uts.edu.au (J.A.); 4Faculty of Medicine and Health, School of Pharmacy, The University of Sydney, Sydney NSW 2006, Australia; 5Department of Research and Clinical Epidemiology, The Canadian College of Naturopathic Medicine, Toronto, ON M2K1E2, Canada; 6Transitional Doctorate Department, Pacific College of Oriental Medicine, San Diego, CA 92108, USA

**Keywords:** naturopathy, prevention, complementary medicine, chronic disease management, policy implications

## Abstract

*Background and Objective*: The concept of a “disruptive innovation,” recently extended to health care, refers to an emerging technology that represents a new market force combined with a new value system, that eventually displaces some, or all, of the current leading “stakeholders, products and strategic alliances.” Naturopathy is a distinct system of traditional and complementary medicine recognized by the World Health Organization (WHO), emerging as a model of primary care. The objective here is to describe Naturopathy in the context of the criteria for a disruptive innovation. *Methods*: An evidence synthesis was conducted to evaluate Naturopathy as a potentially disruptive technology according to the defining criteria established by leading economists and health technology experts: (1) The innovation must cure disease; (2) must transform the way medicine is practiced; or (3) have an impact that could be disruptive or sustaining, depending on how it is integrated into the current healthcare marketplace. *Results*: The fact that Naturopathy de-emphasizes prescription drug and surgical interventions in favor of nonpharmacological health promotion and self-care could disrupt the present economic model that fuels health care costs. The patient-centered orientation of Naturopathy, combined with an emphasis on preventive behaviors and popular complementary and integrative health services like natural products, mind and body therapies, and other therapies not widely represented in current primary care models increase the likelihood for disruption. *Conclusions*: Because of its patient-centered approach and emphasis on prevention, naturopathy may disrupt or remain a durable presence in healthcare delivery depending on policymaker decisions.

## 1. Introduction

The United States health care system is moderately effective in prolonging life, albeit at high costs [1]. There is much room for improvement in health care delivery, and within developed countries, there is near universal agreement on aspirations to achieve the “triple aim” of health care [2]—improved health and patient experience at a lower cost. Multiple tensions exist between these three aims, which render them difficult to achieve. Trade-offs between the delivery of the clinical services patients want versus those services known to reduce costs is an example of one fundamental tension [3]. Additional contextual tensions preventing achievement of these combined aims include the unpopularity amongst patients for approaches that limit access and coverage for services [4,5], and the unpopularity amongst providers for approaches that reduce insurance reimbursement [5]. Technological “innovations” (e.g., electronic health records, gene-based therapies, and advanced imaging technologies) are perceived as holding great promise for achieving this triad of health care aims, yet, in most cases, these innovations have increased costs and fail to improve clinical outcomes [6,7]. This context of moderate outcomes and high-costs with dissatisfied stakeholders has created an opportunity for “disruptive” innovations to emerge in health care delivery [8]. One such disruptive innovation with the potential to transform health care is an effective model of primary care less dependent on costly drugs and technological intervention, one which also explicitly promotes improvements in health.

The concept of a “disruptive innovation” was introduced to the business community by the Christensen Institute in respect to an emerging technology representing a new market force combined with a new value system that eventually displaces some, or all, of the current leading “stakeholders, products and strategic alliances” [9]. This concept has more recently been extended to health care [6,9,10]. In order for a new technology to be considered a “disruptive innovation” in health care, the technology must meet at least one of the following criteria: (1) It must cure disease; (2) transform the way medicine is practiced; and/or (3) have an impact that could be disruptive or sustaining, depending on how it is integrated into the current healthcare marketplace. Examples of candidate disruptive innovations in health care are few, and none have yet proven to be fully disruptive, (e.g., gene-targeted therapies and electronic health records have unmet promise) [8]. Little research or discussion has considered innovative models of health care delivery or a re-shuffling of therapeutic approaches to those with lower costs as potential disruptors.

Naturopathy is a distinct system of traditional and complementary medicine (T and CM) recognized by the World Health Organization (WHO) [11]. The educational model for naturopathy is similar to that in allopathic medical training with its foundation in biomedical physiology and diagnostics. However a unique attribute of naturopathy is a reprioritization of the order of therapeutics (Figure 1), first emphasizing lifestyle-oriented self-care, preventive behaviors, nutrition, physical activity, and stress-management counseling before moving on to clinical nutrition (i.e., targeting pharmacologic actions by nutrients for specific diseases irrespective of nutrient status), herbal medicine, homeopathy, and hands-on manual therapies—all rather than emphasizing over-the-counter and prescription drug therapies or surgical interventions [12,13,14,15,16,17,18,19,20]. There are eight accredited colleges of naturopathy in North America, all of which utilize a primary care foundation of medical education, including outpatient pharmacy and pharmacology training [21,22,23,24]. In the United States, naturopathy is licensed in 20 states, the Washington District of Columbia (D.C.) and the territories of Puerto Rico and the U.S. Virgin Islands [23,25]. Third-party insurance coverage for naturopathy varies by state jurisdiction and ranges from minimal coverage (e.g., California) to legislatively mandated coverage (e.g., Washington State) [23,24]. Select states (e.g., Washington, Oregon, and Connecticut) include some reimbursement to naturopathic doctors through Medicaid, and naturopathic doctors practice in established community-care organizations and/or federally-qualified health centers.

An evidence-informed argument is made below that naturopathy is, or has the potential to become, a disruptive innovation in health care in the United States by meeting all three of the Christensen criteria. Table 1 compares potential disruptive innovations from naturopathy and current conventional biomedicine. As outlined below, a “naturopathic” model of primary care—focused on self-care, the minimal use of costly prescription drugs, and being patient-centered—is a testing ground for achieving the “triple aim” in health care and ultimately disrupting the status quo of the health system.

## 2. Criterion 1: A Disruptive Innovation Must Cure Disease

One of three potential criteria an innovation in healthcare must meet to be considered “disruptive” is to cure disease. Though no clinical trials have been performed evaluating changes in disease incidence from naturopathic practice or individual treatments, and naturopathy thus does not meet this criterion as originally articulated, there are plentiful examples of therapeutics commonly applied in naturopathic practice leading to robust changes in clinical status that could alter the course of “incurable” chronic disease. Important for the potential disruption is that, in many cases, the costs for the therapies recommended are outside of the prescriptive drug model and typically include self-care and treatments classified as “dietary supplements” that are thus sold over the counter or online. The disruption to health care is similar as to a cure because the costs previously directed into the system via pharmaceutical sales are diverted to other businesses, leading to lost revenue to pharmaceutical companies and insurance-based care delivery centers. Dietary supplements and self-care practices already account for an estimated $12.8 billion and $2.7 billion, respectively, for out-of-pocket health care expenditures per year [26]. To illustrate this point, we will discuss two costly chronic diseases significantly improved by therapies typically included in naturopathic practice: Depression and type 2 diabetes.

Depression is another chronic condition that is not well treated by currently available drugs, that drastically reduces health-related quality of life, and that carries an estimated economic cost in excess of US $210 billion annually, with 50% of these costs attributable to work place losses [27]. Yoga and St. John’s Wort, two typical naturopathic approaches for depression, have been extensively evaluated for depression and meta-analyzed [28,29,30,31]. Findings for St. John’s Wort have specifically concluded that it is effective, comparable to drug therapy, and safer than drug therapy [29,30,31]. In addition to positive meta-analyses, yoga has been specifically evaluated in the workplace and appears efficacious for depression [32], including in employees with chronic pain [33]. Thus, by offering guidance on self-care and non-drug interventions, naturopathic practice models have the potential to disrupt the current treatment paradigm and to reduce reimbursement for other primary care, pharmacy and mental health services.

Type 2 diabetes is a costly drain on the health care system, with estimates from the United States suggesting diabetes and its complications costs in excess of $400 billion per year, or approximately one of every three health care dollars spent [34]. Notably, therapies typically recommended in naturopathic practice [14,17], including yoga [35,36], omega-3 fatty acids [37], cinnamon [38,39], chromium [40,41], and carbohydrate reduction [42], all have robust clinical trial and/or meta-analysis support for efficacy in reducing clinical risk factors in type 2 diabetes. Additionally, composite outcomes from naturopathic practice [14] and demonstration projects [17] have led to sustained improvements in glucose control and self-care behaviors, similar to intensive interventions tested in clinical trials [43]. If scaled effectively, shifting therapies toward these out-of-pocket expenditures would disrupt pharmaceutical company revenues, which account for the highest proportional cost of all diabetes services [34].

Research to date has not evaluated a “cure” as an outcome from naturopathic practice, and in fact, very few “cures” exist in medicine at all. While curing disease (and arguably preventing or correcting the course of a chronic disease) would be disruptive, redirecting potential revenue away from service delivery centers, pharmacies and pharmaceutical companies generally is equivalently disruptive and far more achievable. Naturopathic practice represents an effort to improve patient outcomes and redirect the course of chronic diseases, and, as an extension, they could become disruptive if allowed to scale and operate independently of the current system (see Criterion 3 below).

## 3. Criterion 2: A Disruptive Innovation Must Transform Elements of Clinical Practice

A second criteria an innovation in health care must meet to be considered “disruptive” is that it must transform elements of clinical practice [6]. Naturopathy transforms clinical practice by modeling a patient-centered approach, which manifests as longer patient visits, the delivery of pragmatic self-care recommendations, the inclusion of non-drug/non-surgical options, and a strong consideration of patient preferences in care [23,44,45]. The end result is consumer-oriented and includes lateral decision-making through the creation of a functional patient–provider team (in contrast to the current top-down approach that leaves the patient with a low locus of control over their care). Notably, improved physician–patient communications impact adherence to evidence-based practice recommendations, improve patient outcomes in heart disease [46], and improve the delivery of preventive services in primary care [47]; thus, care delivery models that encourage stronger physician–patient relationships are worthy of attention by both insurers and health policy makers because of their potential to reduce costs.

In addition to the form of a typical naturopathic patient visit, the function of a typical visit also varies considerably from allopathic care. Despite similarities between naturopathy and biomedicine in the basic structure of the clinical intake and interview format (i.e., a history of present illness including its attributes, review of systems, medical history, diagnosis and prognosis, etc.), the reach of a typical interview also extends to contributory lifestyle factors, such as physical activity history, dietary pattern, stress management; psychosocial stress, including social determinants of health; detailed medication and dietary supplements reviews; a consideration of relevant environmental exposures at work and at home; clinical prevention strategies including vaccination history, screenings and use of personal protective equipment; and, often, an extensive sleep and digestive history. Fundamentally, the goal of many naturopathic care visits is to identify behaviors, self-care strategies, and modifiable environmental and contextual factors that patients can change in order to improve their health and/or function and reduce their dependency on drugs and higher-force interventions [44,48].

Existing observational practice outcomes and clinical trial data support the efficacy of naturopathy for multiple high morbidity chronic health conditions including: cardiovascular risk, hypertension, depression, anxiety, low back pain, and type 2 diabetes [14,15,17,18,19,49,50]. Of note, none of the clinical studies of naturopathy cited included the use of prescription drug therapies in the tested intervention. Changes of this type, i.e., the effective management of chronic conditions without intensified drug therapy, are contrary to most current medical standards of care for non-communicable diseases [51,52,53,54].

Because type 2 diabetes and other chronic diseases such as cancer, chronic pulmonary diseases, and kidney disease have a fundamental behavioral component, a model of clinical training and practice with the potential to modify patient behavior and inflect the course of patients’ prognosis has great potential to disrupt the status quo. In the context of type 2 diabetes, changes in clinical risk factor control during naturopathic care are accompanied by important improvements in patient mood and self-efficacy for change, with concurrent improvements in self-care, including glucose monitoring, dietary change, and physical activity [17]. Depression frequently co-occurs with diabetes [55] and is an established barrier to improved self-care and medication adherence [56]. Therefore, impacting mood and self-efficacy, with parallel improvements in self-care strategies with established mortality benefits [57,58,59], holds great promise to influence the long-term disease trajectory of type 2 diabetes.

Naturopathy is both traditional and innovative in its application of evidence. Though critics often state there is “no evidence” or too limited evidence for naturopathy, the reality is much more nuanced, and, in many cases, this criticism is not valid. The evidence for naturopathy is not limited to trials of individual drugs; rather, it is sourced from multiple disciplines including clinical trials of individual nutrients, dietary interventions, botanical medicines, mind–body practices, and manual medicine and rehabilitation interventions, with evolving practice models steeped in behavioral sciences [23]. In addition, naturopathy derives evidence from the traditional and historical use of practices like botanical medicine, homeopathy, and hydrotherapy, which, similar to the implementation of many surgical interventions, preceded the advance of evidence-based medicine paradigms. Though case reports, case series, and expert opinions are considered “weak” or “low quality” evidence, the accumulation of case reports and Delphi-process guided consensus statements have made important contributions to the foundation of an evidence base, as well as clinical guidelines. One specific example in primary care is the Eighth Joint National Committee (JNC-8) hypertension management guideline, in which six of the nine core recommendations rely on expert opinion as their evidence base [54].

Delays in the translation of new research into practice is a barrier to improvements in clinical care that has been recognized by the National Institutes of Health [60,61]. In some cases, naturopathic doctors apply evidence that may not otherwise be translated. For example, the United States Preventive Services Task Force (USPSTF) recommends moderate-intensity face-to-face diet and lifestyle counseling by primary care providers to those patients at risk for cardiovascular disease with Grade B evidence [62], yet national statistics are very poor for the delivery of this advice in conventional biomedicine [63,64,65,66,67]. Conversely, recommendations consistent with USPSTF guidance are evident in most patient visits with naturopathic doctors [15,16,19]. Thus, the inclusion of naturopathic care may assist in the delivery of evidence-based preventive services and can improve quality assessments by helping more facilities meet preventive care guidelines. Furthermore, it is constructive to consider the virtues of naturopathy as a nimble practice model for the more rapid translation of evidence into practice, especially evidence related to dietary, nutrient, physical activity, and herbal treatments, plus evidence of harms of prescription drugs.

Naturopathic doctors are also much less likely to use prescription medications as treatment, even when these drugs are included in their scope of practice [23]. Because of differences in their application of therapies and a philosophical aversion to prescription drug therapy in practice, naturopathic doctors are positioned to be more discerning with their use of prescription drugs, more likely to follow standard of care guidelines for the use of drugs, and thus less likely to contribute to unnecessary prescribing (e.g., antibiotics for viral upper respiratory tract infections [68]), therefore potentially reducing iatrogenic causes of illness or death. Notably, most harms of medications are not discovered during Food and Drug Administration (FDA)-regulated phase three clinical trials (often upheld as the gold standard for evidence) but rather are discovered through post-approval epidemiologic surveillance (e.g., in 2014 there were 528,192 new case reports of a serious or fatal outcome from FDA-approved prescription or over-the-counter drugs [69]), which is heavily dependent on provider and patient reporting, with the costs of identified harms passed directly on to insurers and patients [70]. “De-implementation” research is gaining importance as the harms associated with prescription drugs are identified and centers attempt to eliminate unproven procedures and practices [71,72,73,74]. Given iatrogenic errors are now considered the third largest cause of death in the United States [75], additional discretion in the use of prescription drugs is long overdue and is exemplified by naturopathic practice.

These deviations, i.e., less reliance on prescription drugs and reduced referrals to specialists, disrupt the business models of hospitals, large group specialty practices, surgical centers, and pharmacies, and they shift health care expenditures to other “wellness” services, thus contributing to multiple emerging markets.

## 4. Criterion 3: A Disruptive Innovation Will Disrupt or Sustain Depending on Its Integration

The third criterion to be met for a technology to be considered “disruptive” in health care is not a criterion of the technology but rather a criterion that is determined by the reaction of the current market/technology to the candidate disruptor (i.e., whether or not the technology disrupts depends on whether or not it is incorporated into the status quo). Naturopathy has been at a crossroads with “mainstream” conventional biomedicine for hundreds of years, with evidence and arguments advocating for a relationship of opposition, integration or medical pluralism [76,77].

Present reactions to naturopathy within biomedicine suggest there is resistance to its articulation within current care delivery models, and instead a reaction of simultaneous marginalization (by not including naturopathy in most medical centers or insurances) and co-optation (by adopting practices) appears more common. The invention of “integrative medicine” illustrates this point. Though the definition of integrative medicine varies (common definitions of “integrative medicine” include: conventional providers who adopt complementary therapies into their practices; care models that include both conventional and complementary care disciplines offered by different providers in one setting; and/or coordination between providers of different disciplines, including complementary medicine), in this context, it generally includes a merging of conventional care and complementary care. The creation of a recognized integrative medicine specialty board qualification exclusive to medical doctors can be interpreted as a reaction to the possible threat of naturopathy redirecting patients (and revenue) from conventional care centers. Notably, the philosophy of “integrative medicine” [78] is nearly identical to that of naturopathy, despite its more recent creation [45,79,80].

Other evidence suggesting conventional biomedicine will resist the inclusion of naturopathy includes positions by major medical societies and allopathic physician advocacy foundations to prioritize opposition to naturopathy licensure and scope expansion [81,82]. However, this strategy is already proving ineffective. Limiting the geographic range of naturopathy licensure simply directs consumers across state lines and already contributes to an emerging medical tourism market on the west coast of the United States [83].

Implying a requisite separation between science and naturopathy, published editorials refuse to label an intervention for cardiovascular risk reduction derived of naturopathic care that designed and implemented by naturopathic doctors as “naturopathic,” instead insisting it must be labeled “science-based lifestyle advice” [84]. Additional examples include editorials claiming the nutritional guidance offered in naturopathic practice as being the unique role of dieticians [85] and conventional medical news outlets haranguing naturopathy as pseudoscientific and dangerous without providing specific examples [86,87]. These examples suggest opponents are emphasizing scope of practice protection rather than embracing cooperation around a shared goal to improve patient outcomes.

The insurance reimbursement of naturopathy is also a strong determinant in the “integration” that may contribute to naturopathy’s fate. By controlling provider reimbursement and regulating the delivery of services, insurance credentialing and reimbursement for naturopathy may actually stifle its growth, compared to more disruptive cash-based, fee-for-service models or health insurance co-ops leveraging long-term investments in prevention.

Ironically, a fundamental method to ensure naturopathy does not evolve as a disruptive technology is to embrace the inclusion of naturopathic care in current models of care delivery rather than resist or attempt to prevent its inclusion. Not unlike the current leaders in the hybrid car market (e.g., Toyota), a delay in the adoption of hybrid and electric car technologies by other vendors (e.g., US car makers) enabled them to establish their brand and grow their customer base, leading directly to the establishment of a dominant market share (e.g., the Prius**^®^**) and, thus, the power to heavily influence the future of the market. Extending this analogy back to naturopathy, a failure by health care vendors to embrace the public’s growing interest in naturopathy and complementary medicine generally is allowing naturopathy to gain market share by delivering a consumer-oriented model of primary care that is mostly external to mainstream medical centers. To illustrate the potential impact of the inclusion vs. exclusion of naturopathy from the current market, consider a hospital center that includes naturopathy compared to one that does not. The hospital that includes naturopathy can create quality standards around provider credentialing; can place limits the scope of practice, including orders and prescribing; can limit indications for referrals; and can control the overall patient experience within the health center. Maintaining clinical services in-house also increases the likelihood of internal referrals to specialty care. However, a hospital center that does not include naturopathy faces competition from community-based naturopathic clinics, which may provide a highly variable experience in the quality of treatment and care, as well as re-direct potential revenue outside of the hospital. Surveys in Washington State that compared patient satisfaction provided a specific example of naturopathic medical centers outperforming nearly all biomedical care facilities [88]. These community-based naturopathic clinics operate beyond hospital control and are much less likely to route customers to the hospital for specialty care or other clinical services. This scenario also reinforces aspects of Criterion 2; by transforming the care experience and challenging elements of the typical care delivery process, other providers’ clinical practices are disrupted with subsequent implications for inter-professional collaboration and team-based care, as has been seen in other settings [89].

Many characteristics of naturopathy appear to mirror those characteristics sought by the public in their health care [88]; therefore, the survival and growth of naturopathy in the healthcare market appears highly likely. However, the degree and rate of naturopathy’s disruption will also depend on the availability of new research that demonstrates effectiveness and patient-centeredness, increased public awareness of their choices in prevention-based primary care and natural health specialty services, increased access to naturopathy, and whether there is robust integration into current health care “technology.”

## 5. Disruptive Innovations Leverage New Values, and Create New Markets

Leveraging new values in a marketplace is only one aspect of a truly disruptive innovation; another aspect is the creation of (or contribution to) new markets. Naturopathy supports emerging markets in several ways, including supporting the development of “wellness”-based economic models of health care delivery through referrals to clinical providers not yet incorporated into mainstream health centers (e.g., massage therapists, yoga therapists, acupuncturists, and health coaches), recommendations for health products not widely available in mainstream pharmacies, clinics and hospitals (e.g., dietary supplements, homeopathic remedies, herbal medicines, and natural health products), and the use of exercise and other movement-based facilities not widely accessible via mainstream rehabilitation centers (e.g., gyms, health spas, and yoga studios). In the United States alone, the out-of-pocket expenditures on complementary and integrative health services exceed $30.2 billion [26]. An additional contribution to the “wellness” economy, albeit not easily quantifiable, is the contribution of naturopathy to the food industry through guidance toward organic, natural foods. The recent purchase of Whole Foods natural food markets by Amazon, Inc. for $13.7 billion speaks to the growing economic force and market share held by the natural food industry [90].

While these statistics are largely derived from United States, there is reason to believe the disruption caused by naturopathy is not limited to the United States, or North America. Public use of and out-of-pocket expenditures for complementary and alternative health services have increased globally [91]. In Australia, for example, available data suggest at least 44.1% of Australians visited a complementary medicine practitioner within the previous 12 months, with similar numbers of visits to complementary and alternative medicine practitioners compared to conventional medical practitioners (69.3 million), with an estimated annual expenditure of $4.13 billion Australian dollars (US $3.12 billion) [92].

Simply redirecting some proportion of the public’s available out-of-pocket health care expenditures toward a different segment of health care providers and products alone does not create a viable market—some measurable benefit or effect of this redistribution must also be created. An emerging case exists that not only do expenditures toward naturopathic care prove cost-effective, the contributions of naturopathy may constitute a “value added” to the workplace. For example, a clinical trial of naturopathy for primary cardiovascular disease prevention demonstrated effectiveness and cost-effectiveness, with a decisive contribution to cost savings being reductions in work place presenteeism (a confusing term, “presenteeism” is working while sick or with otherwise reduced physical and/or emotional function [93]), not the direct costs of the care provided [94]. Some authors have extended the potential of the value added by naturopathy to posit it has greater potential to reduce escalating costs in the developed and developing world due to its emphasis on prevention and patient self-management in the context of an escalating prevalence of non-communicable diseases [95].

Some critics have argued that the additional expenditures on complementary and integrative health services are a “double-dip” by the public (i.e., many people use complementary medicine providers concurrent to their primary care providers), leading to excess medical spending. Though there is limited evidence available on this topic, in states where naturopathic doctors are licensed as primary care providers (e.g., Washington state), over utilization does not appear to occur and people who use complementary medicine providers tend to be in better health, have fewer diagnostic tests and attend fewer costly specialty visits [23].

Perhaps the greatest potential for disrupting the dominant market force in medicine is the ephemeral promise of chronic disease prevention through the routine delivery of primary prevention services and by optimizing and scaling behavior change strategies. Historically, investment in prevention by third-party payers has been limited by the expectation of a five-year return on investment (ROI), which simply is not possible for prevention strategies targeting chronic disease, from which the ROI is likely 20 or 30 years away from the point(s) of intervention. Again, type 2 diabetes provides an illustrative case example for this point. The individual level costs of type 2 diabetes are estimated at $85,200.00 per person, translating into an approximate cost of $1.7 trillion for the 20 million adults in the United States with type 2 diabetes (not accounting for increasing costs of treatment, disease prevalence, or the economic losses from increased workplace absenteeism and presenteeism) [96]. Thus, the redirection of even 1% of the direct costs of diabetes care toward primary prevention-oriented clinical services would create a new US$ 17 billion market.

While naturopathy has not demonstrated the ability to “cure” or “prevent” type 2 diabetes in clinical trials, the practices recommended during naturopathic consultations for type 2 diabetes are consistent with the recommendations in the Diabetes Prevention Program [97], as they engage on multiple behavioral barriers to lifestyle change, translate into improved self-care, and appear to lower blood glucose [14,17]. Combining those changes with reduced primary cardiovascular disease risk [19] may create a significant inflection point in a patients’ health/risk trajectory for multiple chronic diseases. Additionally, although evidence is still quite limited, at least two herbal medicine products have clinical trial evidence for diabetes prevention [98,99], and numerous nutrient or herbal interventions used in naturopathic practice have evidence for lowering blood glucose or otherwise impacting diabetes-related risk [100,101]. Thus, naturopathy offers several desirable health products to the savvy consumer, including multiple categories of evidence-based therapeutic interventions not routinely delivered in conventional biomedicine, with overarching goals of primary prevention and functional health improvement—all of which hold great promise for creating inflection points in the health trajectory for people with diabetes.

## 6. Conclusions and Future Directions

By meeting all three criteria outlined by the Christensen Institute, naturopathy appears to constitute a disruptive innovation in health care. If optimized, the model of health care represented by naturopathy has the potential to surpass allopathic care (in its current form) as a dominant model of primary care by representing patient- and family-centered values in care delivery, a focus on prevention, and the de-emphasis on pharmaceuticals and elective surgery in favor of pragmatic self-care behaviors, lifestyle practices, and natural treatments. Notably, promoting healthy self-care behaviors, de-emphasizing costly technologies that do not improve outcomes, and becoming more judicious in the use of prescription drugs are all known opportunities to achieve the triple aim [102]. Thus, naturopathy may contribute to wider-ranging health service goals. Whether these changes occur due to market pressures encouraging biomedical care providers and academic training centers to change their model or due to naturopathic doctors playing an increasing role in care delivery remains uncertain. Given the many barriers to rapid change in allopathic models of training and care delivery, reductions in public trust of managed care [103,104], and an increasing number of deaths attributable to medical errors, perhaps it is time to embrace the approach represented by naturopathy and actively seek to incorporate it more widely rather than marginalize its role. As common diagnostic and treatment algorithms become increasingly accessible through mobile applications and artificial intelligence, the survival of biomedical primary care may require it to redefine itself and mirror the values, if not the interventions, represented by modern naturopathy.

## Figures and Tables

**Figure 1 medicina-55-00603-f001:**
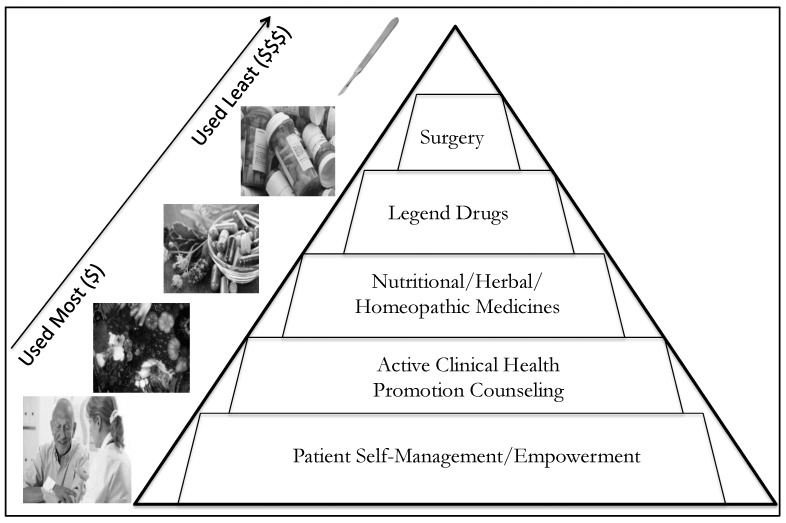
Hierarchy of typical therapeutics applied in naturopathy practice.

**Table 1 medicina-55-00603-t001:** Comparison of potential disruptive technologies in naturopathic and allopathic primary care.

Criterion	Allopathy/Conventional Biomedicine	Naturopathy/Naturopathic Medicine
Cure disease?	Genetic medicine, biologic therapies, gene editing, robotic surgery, precision radiation	Chronic disease prevention, self-care and “out-of-pocket” treatments
Transform practice?	Electronic Health Records, medical homes, evidence-based practice	Patient-centered, prioritization of patient preferences, experts in complementary medicine
Disrupt or Integrate?	Integrated via guidelines and standards of practice upon reaching a threshold of evidence and provider acceptance	Integrated through insurance coverage, “integrative” guideline development and co-location of providers OR Disruptive via marginalization but increased consumer demand and market pressures

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
