# Peer review of "Naturopathy as a Model of Prevention-Oriented, Patient-Centered Primary Care: A Disruptive Innovation in Health Care"

_medicina, 2019, doi:10.3390/medicina55090603_

Round 1

Reviewer 1 Report

The commentary article, “Naturopathy as a Model of Patient-centered, Prevention-oriented Primary care: A Disruptive Innovation in Health Care?” debated the naturopathy using disruptive innovation theory, discussed using comment from Christensen Institute, and extended to health care. Three criteria: 1) the innovation must cure disease; 2) must transform the way medicine is practiced; or 3) have an impact that could be disruptive or sustaining, depending on how it is integrated into the current health care marketplace.

I think the first criteria cannot meet using naturopathy in health care, though the authors extended the first criteria “cure disease” to also include prevention, cure, or other significant inflections in the prognosis of disease.

The authors used type 2 diabetes, and other chronic diseases such as cancer, chronic pulmonary diseases … to expand “cure” was a wrong way. It is not about prevention or to change the predicted course of diseases.

They may put the argument of “wellness promotion” and “public health” on criteria 2 or even criteria 3.

Thus, I suggest the manuscript to rearrange the contents, the authors need to think and offer a good example on criteria 1 “cure diseases”. The current argument on page 4, lines 113-122, would change to criteria 2 or criteria 3.

If the authors change this part of argument, then the following discussion will be modified based on the debates.

Author Response

Dear Reviewers and Editors:

We are thankful for the opportunity to respond to you regarding feedback on our recently submitted manuscript titled “Naturopathy as a Model of Prevention-oriented, Patient-centered Primary care: A Disruptive Innovation in Health Care”. In response to your critique we have carefully prepared a response, and have resubmitted a heavily re-organized manuscript. We are hopeful you’ll find our commentary to be of interest, and we anticipate it will generate considerable discussion and commentary from your readership. Our commentary is heavily cited in recent literature in order to accurately represent the present status of naturopathy, rather then promote historic biases. Due to the heavy revisions to the manuscript, we have enclosed a “tracked” and “clean” version. Please let us know if there are any difficulties in resolving the edits made in response to your feedback. We appreciate the additional opportunity to response to the Reviewers, point-by-point below:

Reviewer #1:

Comment: Argument does not meet the intention of Criterion 1, the requirement to “cure” disease, and the current discussion should be re-organized under Criterion 2, or Criterion 3.

Response: Thank you for your thoughtful review and suggestion. We acknowledge the absence of evidence do data on cure from naturopathy or its practices. We have acknowledged this fact more clearly and directly in the revised manuscript. Based on your feedback, we also moved the bulk of the contents under the discussion of Criterion 2.

Based on feedback from Review #2, who asked for more discussion of highly evidence-supported therapies in naturopathic practice, we do maintain the potential for disruption remains when out-of-pocket therapies are substituted, with benefit, for reimbursed costs, because of the billions of dollars in potential revenue that are no longer received by care delivery organizations, pharmacies and pharmaceutical companies. Two case examples, depression and diabetes, are now included with additional details.

The authors respectfully point out meeting all 3 Criterion is not required for a technology to be considered “disruptive” thus we are hopeful the changes in organization and the modified commentary now included will adequately support the overall argument.

Reviewer 2 Report

Although this is intended to be a commentary, several areas are grossly underdeveloped and require greater clarity and substantiation.

The 'triple aims' of health care was stated incompletely. It should be "improving the patient experience of care (including quality and satisfaction), improving the health of populations, and reducing the per capita cost of health care." "The educational model for naturopathy is similar to that in allopathic medical training with its foundation in biomedical physiology and diagnostics, with a change in the order of therapeutics". Much of the premise of the current argument relies on the fact that naturopathy is evidence-based and beneficial system of practice. It is important to provide a well-rounded debate by considering the fact that many mainstream health care professionals still find naturopathy pseudoscientific at best, and the profession of naturopathic medicine remains significantly fragmented (citation: ncbi.nlm.nih.gov/pubmed/15130568). Naturopaths claim that they practice based on scientific principles. Yet examinations of naturopathic literature, practices and statements suggest a more ambivalent attitude; a review of the therapies advertised on the websites of clinics offering naturopathic treatments does not support the proposition that naturopathic medicine is a science and evidence-based practice (citation: ncbi.nlm.nih.gov/pmc/articles/PMC3182944). The central belief, “vitalism”, posits that living beings have a “life force” not found in inanimate objects. Vitalism as a concept was disproved by Wöhler in 1828, yet the idea continues to thrive in naturopathy. Naturopathic treatment ideas are all grounded in the idea of restoring this “energy”, rather than being based on objective science. The lack of governing bodies and authorities in several developed countries even, mean that there are practitioners who prescribe combinations of vitamins, minerals, and herbs at doses that would cause toxicity, as well as naturopaths who advocate for the long-discredited blood type diet (citation: ncbi.nlm.nih.gov/pubmed/17377563). Although the authors assert that this is an "evidence-based review", there is a paucity of evidence or referencing to clinical studies to support naturopathic interventions. It would be more useful to have a table summarizing the evidence-based interventions as well as the supporting clinical studies to make the case for naturopathy. For example, St John's wort (citation: ncbi.nlm.nih.gov/pubmed/28064110) and Saffron (citation: ncbi.nlm.nih.gov/pubmed/30036891) may be efficacious patients with mild-to-moderate depression. Curcumin, on the other hand, has anti-inflammatory properties and has demonstrated benefits for patients with irritable bowel syndrome (citation: ncbi.nlm.nih.gov/pubmed/30248988). Authors could also mention the WHO’s Mental Health Gap Action Program (mhGAP), which is a solution proposed to tackle the shortage of mental health specialists in rural areas. This program aims to increase mental health providers by recruiting health workers who are not specialist in mental health; the wide distribution of Traditional and Complementary Medicine (TCM) practitioners in lower-income countries make them an appealing candidate. Incorporating TCM into conventional medicine to treat depression has various benefits such as better accessibility, acceptability and lesser stigma (citation: ncbi.nlm.nih.gov/pmc/articles/PMC4456435/). Besides the "double-dip" issue, the cost-effectiveness of CAM therapies cannot be properly evaluated without first establishing its effectiveness through rigorous studies. For example, spinal manipulative therapy for back pain may offer cost savings to society, but it does not save money for the purchaser. There is a paucity of rigorous studies that could provide conclusive evidence of differences in costs and outcomes between other complementary therapies and orthodox medicine. The evidence from methodologically flawed studies is contradicted by more rigorous studies, and there is a need for high quality investigations of the costs and benefits of complementary medicine (citation: ncbi.nlm.nih.gov/pubmed/10859604). Please change "Next Directions" to "Future Directions". Please change "increasing healthy self-care" to "promoting healthy self-care". Patient-centred care has multiple aspects and it is often assumed to be superior to a paternalistic approach in the management of chronic diseases, however, this hypothesis has not been rigorously tested or supported by current evidence. Improved communication between patients and providers, and continuity of care are associated with increased provision of preventive services, while other aspects of patient-centred care are not strongly related to the effective delivery of preventive services (citation: ncbi.nlm.nih.gov/pmc/articles/PMC1492576). Patient-centered care is a form of value-based care. Governmental agencies and insurance companies are using these new care models as proxies for quality. Since quality is difficult to measure, a patient’s experience or satisfaction with the care provided serves in its place. Unfortunately, the fallacy of patient-centered or value-based care is that it forces us to allocate resources on an outcome that has little to do with the true quality of care provided. Is the patient a customer, and as a customer, is the patient always right? One has to merely look at the ever-growing opioid epidemic in the US to see the error in that logic.

Author Response

Reviewer #2:

Comment: Triple aim was incompletely stated.

Response: Thank you. The reference to the Triple Aim was removed because the authors agree it was incompletely developed.

Comment: The current argument depends on the premise Naturopathy is evidence-based and effective.

Response: Thank you. The authors agree additional research is necessary, including comparative effectiveness trials. The current manuscript is heavily cited with recent evidence on contemporary naturopathic practice. The practices have evolved considerably over its life course, however the modern, accredited model of naturopathic education, and the licensure in multiple jurisdictions, is as a model of primary care. Research supporting medical visits at naturopathic academic medical centers support this role in care. Fundamental to its consideration as a “disruptive innovation” and in the discussion of Criterion 3, is the emerging nature of naturopathy in primary care, and the question of how established will naturopathy become. If the effectiveness research is performed and demonstrates effectiveness above current primary care standards, will it become more disruptive because of its outcomes, or less disruptive because it gets gradually more incorporated? These questions are inherent in our commentary and we are hopeful the readership will participate in this discussion through their commentary as well.

Comment: Vitalism has been disproven.

Response: Thank you for this commentary. Vitalism is a dated conceptual model that does not represent modern naturopathic training, scope of practice or approaches to care. Modern naturopathic medical education does not endorse vitalism. However, naturopathic medicine is arguably more “vitalistic” in its approach to care, as treatments tend to be supportive of the host, as well as, targeted toward disease. Naturopathy is also reported and discussed as “patient-centered” in qualitative research, however “vitalistic” is not a theme that has ever emerged from qualitative interviews of naturopathic patients, nor evaluations of naturopathic practice, thus we do not discuss vitalism in this commentary piece.

Comment: Naturopathy not advertised as evidence-based.

Response: We’ve limited our commentary to the available peer-reviewed literature regarding outcomes from naturopathy, and descriptions of services provided in key demonstration projects and formal evaluations. Many clinicians, of many backgrounds, advertise trendy clinical services; this phenomenon is not limited to naturopathic providers. Also, of note, although the authors of the citation included by Reviewer 2 does include several therapies with little to no evidence, many of the therapies advertised by the naturopathic clinics under scrutiny were potentially evidence-based, depending on the indication. Therapies such as botanical medicine, clinical nutritional, lifestyle counseling, massage, and acupuncture have available evidence for common primary care conditions. Also, notably, although scrutinized heavily, IV chelation therapy is in clinical trial in type 2 diabetes (i.e., the current TACT2 trial supported by the NIH) because of a robust 48% reduction in mortality in diabetes identified in TACT1 (PMID: 31101487).

Comment: Greater discussion of evidence-based therapies is needed.

Response Thank you. In our revised manuscript, we provide additional evidence for naturopathic therapeutics in the context of care for depression and diabetes. Similar extrapolations could be provided for multiple chronic, non-communicable diseases, but diabetes and depression were chosen due to their high cost and general impact on health services.

Comment: Change “Next Directions” to “Future Directions” and “increased…” to “promoted….”

Response:  Thank you, we have incorporated these recommended edits into the revised draft.

Comment: Improve discussion re: the limitations of “patient-centered care”.

Response:  Thank you. We have aimed to clarify this discussion to be clear we are interested primarily in the disruptive potential of a prevention-based model, and the value improved patient communication may have in achieving that aim, as represented by naturopathic medicine.

We are hopeful you will find our revision suitable for publication in the upcoming special issue of Medicina, focused on Complementary Medicine.

Thank you for this opportunity to respond to reviewers in detail, and we hope this manuscript will be of considerable interest to your readership, if only to provoke continued academic dialogue.

Round 2

Reviewer 1 Report

The revised manuscript seems well. I have no more comment.

Reviewer 2 Report

Thank you for considering the reviewers' comments and making the necessary revisions.